# Knockout of *stim2a* Increases Calcium Oscillations in Neurons and Induces Hyperactive-Like Phenotype in Zebrafish Larvae

**DOI:** 10.3390/ijms21176198

**Published:** 2020-08-27

**Authors:** Rishikesh Kumar Gupta, Iga Wasilewska, Oksana Palchevska, Jacek Kuźnicki

**Affiliations:** International Institute of Molecular and Cell Biology in Warsaw, Ks. Trojdena 4, 02-109 Warsaw, Poland; rkgupta@iimcb.gov.pl (R.K.G.); iwasilewska@iimcb.gov.pl (I.W.); opalchevska@iimcb.gov.pl (O.P.)

**Keywords:** SOCE, Stim2a, calcium toolkit, zebrafish, behavioral tests, hyperactivity, glutamate, PTZ

## Abstract

Stromal interaction molecule (STIM) proteins play a crucial role in store-operated calcium entry (SOCE) as endoplasmic reticulum Ca^2+^ sensors. In neurons, STIM2 was shown to have distinct functions from STIM1. However, its role in brain activity and behavior was not fully elucidated. The present study analyzed behavior in zebrafish (*Danio rerio*) that lacked *stim2a*. The mutant animals had no morphological abnormalities and were fertile. RNA-sequencing revealed alterations of the expression of transcription factor genes and several members of the calcium toolkit. Neuronal Ca^2+^ activity was measured in vivo in neurons that expressed the GCaMP5G sensor. Optic tectum neurons in *stim2a^−/−^* fish had more frequent Ca^2+^ signal oscillations compared with neurons in wildtype (WT) fish. We detected an increase in activity during the visual–motor response test, an increase in thigmotaxis in the open field test, and the disruption of phototaxis in the dark/light preference test in *stim2a^−/−^* mutants compared with WT. Both groups of animals reacted to glutamate and pentylenetetrazol with an increase in activity during the visual–motor response test, with no major differences between groups. Altogether, our results suggest that the hyperactive-like phenotype of *stim2a^−/−^* mutant zebrafish is caused by the dysregulation of Ca^2+^ homeostasis and signaling.

## 1. Introduction

In any cell, the precise regulation of Ca^2+^ flow is possible because of the existence of different branches of calcium signaling that allow Ca^2+^ influx into the cytosol and then into Ca^2+^ depots, including the endoplasmic reticulum (ER), mitochondria, and cytosolic Ca^2+^-binding proteins. One such pathway is store-operated Ca^2+^ entry (SOCE), which is responsible for restoring Ca^2+^ levels in the ER, the largest store of Ca^2+^ [1]. The main proteins that are involved in SOCE are stromal interaction molecule (STIM) proteins, Orai and transient receptor potential (TRP) channels. STIMs are localized to the ER membrane and act as Ca^2+^ sensors via their luminal EF-hand Ca^2+^-binding site. Upon a decrease in Ca^2+^ levels in the ER lumen, the EF-hand changes its conformation, leading to the formation of clusters of STIM molecules and their translocation to the plasma membrane [2,3,4]. In the plasma membrane, STIM proteins form complexes with plasma membrane channels, including Orai (Orai1, Orai2, or Orai3) [5] and TRP (especially members of the TRPC and TRPV families) [6]. Upon these interactions, Ca^2+^ enters the cell to refill ER stores via Sarco/endoplasmic reticulum Ca^2+^-adenosine triphosphatase (SERCA) [5,7,8]. Upon cellular stimulation, the opposite process of Ca^2+^ release from the ER to the cytosol occurs via such channels as ryanodine receptors and inositol 1,4,5-triphosphate 3 receptors [1,9]. Under normal conditions, the level of Ca^2+^ in the cytosol is kept at a very low concentration and increases during cell activation. However, an excessive increase in Ca^2+^ levels activates cascades that lead to cell death [1,9]. The proteins that are involved in Ca^2+^ homeostasis and signaling are encoded by calcium toolkit (CaTK) genes [10,11].

The other branches of Ca^2+^ signaling depend on voltage-gated calcium channels and receptor-operated channels. Among receptor-operated channels that are the best characterized and have been shown to interact with SOCE components are glutamate receptors, α-amino-3-hydroxy-5-methyl-4-iso-xazolepropionic acid receptors [12], and *N*-methyl-D-aspartate receptors [13]. These receptors are highly expressed in neuronal cells where they support neuronal excitability, synaptic plasticity, and neurotransmitter release [1,14,15,16,17,18]. Store-operated calcium entry has been shown to participate in the regulation of other Ca^2+^ signaling pathways (reviewed in [9,19]), including the regulation of such transcriptional networks as the calcineurin/nuclear factor of the activated T-cell signaling cascade, nuclear factor κB [20], Sp4 [21], and adapter protein-1 [22]. Thus, neuronal SOCE can participate in the regulation of neurodevelopmental processes and cell fate via specific transcriptional programs of c-*fos*, *neuronal PAS domain-containing protein-4*, and *endothelial growth factor-1* transcription factors, also known as immediate early genes [13,23,24,25,26]. Moreover, the dysregulation of SOCE was shown to be involved in neurodegenerative diseases and other pathologies of the central nervous system (reviewed in [1,9]).

Store-operated calcium entry mechanisms have been shown to act in neurons similarly to non-excitatory cells [27]. However, the composition of proteins that are involved in this process varies depending on the tissue [28]. STIM2 was shown to have higher expression levels than STIM1 in specific brain regions, such as the hippocampus [29,30] and cortex [27], suggesting an important role of STIM2 in the mouse brain [31]. Several attempts have been made to investigate behavior in *Stim2* knockout mouse lines [31,32]. In Oh-hora et al., the *Stim2* knockout mouse line exhibited growth retardation at 4–5 weeks of postnatal development and did not survive longer than 5 weeks [33]. The reasons for this early lethality and behavioral characteristics were not addressed in this previous study. The mouse line with *Stim2* deletion that was evaluated by Berna-Erro et al. had no visible phenotype, but postnatal lethality was observed at 8 weeks, and only 10% of *Stim2^−/−^* mice reached 30 weeks of age [31]. Differences in the effects of *Stim2* knockout in these mouse lines could be attributable to the different genetic backgrounds of the experimental animals [31]. In *Stim2^−/−^* mice that were younger than 8 weeks of age, cognitive deficits were observed in the Morris water maze compared with wildtype (WT) littermates. This test focuses on spatial memory function, which depends on neuronal function in the hippocampus, a brain region where STIM2 levels are high. No changes in anxiety-like behavior were detected in the elevated plus-maze [31]. The issue of early lethality was overcome by restricting *Stim* knockout only to the forebrain [32]. *Stim2* knockout did not affect anxiety-like behavior. In the water maze test, STIM2-deficient animals performed as well as WT animals, suggesting the preservation of spatial memory ability [32]. Double *Stim1/Stim2* knockout resulted in an increase in activity in the visual cliff test, impairments in spatial memory, and an increase in long-term potentiation [32], indicating that the loss of both STIM1 and STIM2 proteins impairs synaptic transmission. Thus, studies in mice that have been published to date provide interesting but inconclusive results, and further comprehensive analyses of the role of STIM2 in brain activity and behavior are needed.

Zebrafish (*Danio rerio*) is gaining popularity as a model organism for neuroscience research because of its unique features that allow whole-brain imaging in vivo and the similarity of its molecular pathways [34,35] and brain physiology [36] to humans. Moreover, zebrafish are a powerful model organism that deepens our knowledge of brain function and elucidates links between neuronal circuits and behavior [37]. Several behavioral tests have been adopted for zebrafish larvae to characterize molecular determinants of behavior [38,39,40,41,42] and perform drug screening [36,43,44]. Notably, the genetics of *Danio rerio* are well established and allow the creation of knockdown and knockout fish [45]. Importantly, the zebrafish genome contains many human homologues in more than one copy through genome duplication. Genome duplication, however, poses some problems for the creation of complete knockout lines, but such peculiar genetics of zebrafish also have some advantages. For example, partial knockout may prevent lethality. The majority of components of SOCE are present in the zebrafish genome, including *stim1a*, *stim1b*, *stim2a*, *stim2b*, *orai1a*, *orai1b*, and *orai2* [46], the expression of which in the zebrafish brain and heads of larvae was confirmed by our research group [10]. Store-operated calcium entry was studied in zebrafish but only at the early stages of embryo development [47]. Tse et al. reported that SOCE-like activity in neurons derived from dissociated neurospheres of zebrafish [48]. Zebrafish Stim1 was shown to be crucial for the axon guidance of motor neurons and the regulation of calcium signaling [49]. We recently showed that the knockout of one orthologue of *Stim2*, *stim2b*, caused hyperactivity and increased the susceptibility to seizures in zebrafish larvae [50]. Considering that Stim2 isoforms might have diverse effects on brain activity, the role of Stim2a should also be investigated. Thus, the present study sought to complement our previous work with *stim2b* knockout zebrafish and form a complete picture of the role of Stim2a in the brain. We combined several behavioral tests with in vivo calcium imaging to reveal that changes in neuronal activity affect behavior in *stim2a*^−/−^ animals. We also identified some candidate genes that could be responsible for such changes.

## 2. Results

### 2.1. Genotyping of stim2a Knockout Zebrafish Line and Phenotypic Analysis

We generated the *stim2a^−/−^* zebrafish line using CRISP/Cas9 technology. The deletion of 5 bp in front of the SOAR domain sequence produced an early stop codon. This resulted in the truncation of the target protein at 318 amino acid residues (Figure 1A,B). Animals with the mutation were identified by genotyping DNA that was isolated from fin tissue using a restriction length fragment polymorphism (RLFP). The 5 bp fragment that is recognized by the MBOII restriction enzyme was absent in *stim2a^−/−^* zebrafish (Figure 1B). Larvae from mutant animals did not exhibit developmental abnormalities and exhibited no changes in viability until 120 hpf (Figure 2A). Adult fish were viable and fertile. No visible anatomical changes were detected. Thus, the lack of *stim2a* did not induce a significant morphological phenotype (Figure 2B).

### 2.2. Higher Thigmotaxis and Hyperactivity in the Open Field Test in stim2a^−/−^ Larvae

To investigate behavior in *stim2a^−/−^* fish, we evaluated thigmotaxis in the open field test using 4 dpf old larvae (Figure 3A). Thigmotaxis reflects the preference of animals for the edge of the well compared with the central area [41]. The experiment was performed for 15 min. The data analysis was divided into three phases, 5 min each. During the first phase, the animals could be assumed to be stressed because they were transferred from the dish to a novel place. Both WT and mutant animals spent more time at the edge of the well compared with the center, which can be interpreted as anxiety-like behavior (Figure 4B). Mutant fish exhibited higher mobility than WT fish both at the edges and in the center, indicating hyperactivity. During the second phase (5–10 min interval), WT animals began to explore the whole well, manifested by a similar amount of movement in both the center and the edge of the well (Figure 4A). In contrast, mutant fish still preferred swimming at the edge of the well and avoided the center (Figure 4). In the third phase (10–15 min interval), both groups behaved similarly, spending a similar amount of time in the center and at the edge (Figure 4B). Mutant fish reached a higher maximum acceleration (Figure 4C) and spent more time moving at high speed (Appendix A) than the WT during the first 10 min of the experiment. During all phases of the experiment, *stim2a*^−/−^ mutant larvae covered a significantly bigger total distance than WT fish (Figure 4D). *stim2a*^−/−^ mutant larvae moved with a higher velocity (Appendix A) and spent more time moving (Appendix A). These results indicate that *stim2a^−/−^* zebrafish are hyperactive and required more time to habituate to the new environment, reflecting an increase in thigmotaxis.

### 2.3. Lower Phototaxis and Hyperactivity in stim2a^−/−^ Larvae in the Light/Dark Preference Test

We then investigated whether *stim2a^−/−^* zebrafish larvae had different light/dark preference from WT zebrafish, known as phototaxis [51]. Specifically, we analyzed the amount of time the fish spent in the light half and dark half of the Petri dish (Figure 3B). Wildtype fish traveled a longer distance (Figure 5A) and spent more time in the light zone (Figure 5B). In contrast, the *stim2a^−/−^* mutant zebrafish did not exhibit significant phototaxis and swam a comparable distance as the WT fish (Figure 5A) and spent as much time in either zone of the Petri dish (Figure 5B). Another difference between the WT and mutant fish was the overall mobility. The total distance traveled in both zones of the Petri dish was approximately four times higher in the mutant fish than in the WT fish (Figure 5C). These data suggest that *stim2a* deletion led to hyperactivity and lower phototaxis.

### 2.4. The stim2a^−/−^ Zebrafish Larvae Reacted to Light Changes but Exhibited an Increase in Mobility during the Low-Activity Phase

To further characterize the behavioral patterns and check vision in *stim2a^−/−^* mutants, we performed the visual–motor response test (see Figure 3C). This test is based on the ability of zebrafish larvae to react to modulations of light intensity by emitting a startle response, similar to masking phenomena in higher vertebrates [52]. In this test, we measured the long-term changes in their activity that resulted from switching the light and dark phases. We analyzed changes in the total distance traveled during three phases of the experiment: baseline, low activity, and high activity (Figure 6). A decrease in mobility occurred upon the reaction of larvae to the sudden change in light when they were switched to the low activity phase. The zebrafish larvae regained their mobility during the subsequent high activity phase in the dark.

During the baseline phase, we did not detect any difference in the mobility between WT and mutant larvae. Upon the subsequent increase in light intensity, both genotypes exhibited a decrease in mobility. Interestingly, the *stim2a^−/−^* larvae moved a significantly longer distance when the light was turned on. Finally, both WT and mutant zebrafish exhibited an increase in activity during the subsequent high activity phase after turning the light off. However, no significant difference in the total distance traveled was observed between the groups during this phase, similar to the baseline phase (Figure 6). The results of the visual–motor response test further supported the hyperactivity phenotype of *stim2a^−/−^* larvae that was observed in the open field test (Figure 4) and light preference test (Figure 5), in which the mutant fish exhibited hyperactivity upon the abrupt switching of the light. Importantly, the mutants were able to distinguish between the light and dark phases as efficiently as the WT animals, indicating that they were able to react to light.

### 2.5. Calcium Imaging Revealed an Increase in Neuronal Activity in the Brain in stim2a^−/−^ Larvae

We performed in vivo Ca^2+^ imaging to explore the possible changes in neuronal activity in *stim2a*^−/−^ zebrafish larvae. We can clearly see that the brain consists of tightly packed somata (almost no extracellular space) in WT and *stim2a^−/−^* fish (Appendix A). Spontaneous Ca^2+^ oscillations in neurons in WT and *stim2a*^−/−^ larvae were monitored using the GCaMP5G Ca^2+^ sensor. In all the analyzed zebrafish lines, we chose the neuronal cells from the sub-region in the periventricular gray zone of the optic tectum to analyze the spontaneous Ca^2+^ activity as it was showing intensive oscillations and was easily identified in each experiment to allow comparisons between them. The average oscillation frequency was significantly higher in *stim2a^−/−^* fish compared with WT (Figure 7A), which indicates that neurons were firing with higher frequency in *stim2a^−/−^* larvae because of an increase in neuronal activity in this brain region. However, the average Ca^2+^ amplitude was significantly low in *stim2a^−/−^* mutants as compared with WT (Figure 7B). After that, we checked the changes in the Ca^2+^ oscillation frequency (Figure 7C) and average Ca^2+^ amplitude (Figure 7D) after 600 µM glutamate treatment WT and *stim2a*^−/−^ larvae (Appendix A). We observed a significant increase in the average Ca^2+^ oscillation frequency in the case of the *stim2a^−/−^* mutant. However, the difference in average Ca^2+^ oscillation amplitude between WT and *stim2a*^−/−^ mutant after glutamate treatment was not significant.

### 2.6. Treatment with Pentylenetetrazol (PTZ) and Glutamate Resulted in Minimal Differences between stim2a^−/−^ Mutants and WT in the Visual–Motor Response Test

We further explored the alterations of neuronal activity, detected by in vivo calcium imaging. We administered two drugs that stimulate neuronal activity, PTZ and glutamate, and performed the visual–motor response test (see Figure 3C). Pentylenetetrazol induces seizures, and zebrafish were shown to exhibit behavioral reactions in response to this treatment beginning at 3 dpf [53,54]. Glutamate increases neuronal activity and locomotor activity [50,55]. Importantly, glutamate was shown to penetrate zebrafish larvae and reach brain neurons. Thus, the main effect of this drug on animal behavior is primarily attributable to an increase in brain activity [50,55]. We performed PTZ treatment from a lower to a higher dose [53]. At a lower dose of PTZ, we could not detect any differences in the WT compared to the mutant (Appendix A). However, both the mutant and WT fish reacted to both 15 mM PTZ and 600 μM glutamate treatments, reflected by an increase in the distance traveled during all phases of the experiment (Figure 8). Interestingly, the mutant larvae reacted to 600 μM glutamate with a significantly milder induction of locomotor behavior during the baseline phase (Figure 8B). No difference in the distance traveled was observed between the two genotypes that were exposed to 15 mM PTZ (Figure 8A). Both WT and *stim2a^−/−^* larvae significantly increased their locomotor activity, which was the result of the occurrence of seizure-like episodes. Consistent with the results that are presented in Figure 6, we observed the most pronounced difference between the WT and *stim2a^−/−^* fish during the low activity phase under the no-treatment condition (Figure 8A,B). Both mutant and WT larvae exhibited hyperactivity and no decrease in mobility with 15 mM PTZ treatment in the light (Figure 8A). Finally, we did not observe any difference in the behavior between *stim2a^−/−^* and WT larvae during the high activity phase with either treatment (Figure 8A,B).

### 2.7. RNA Sequencing Revealed Differential Gene Expression in stim2a^−/−^ Zebrafish Larvae

To gain insights into molecular changes that were caused by *stim2a* deletion, we analyzed differential gene expression in the *stim2a* mutant and WT fish using RNA sequencing (Figure 9A, Appendix A). A total of 392 genes showed at least two-fold change (upregulation: 336 genes, ~86%; downregulation: 56, ~14%) in expression (Figure 9B). Using qPCR, we checked the level of expression of SOCE components, but no differential gene expression was found between genotypes, except *stim2a* downregulation (Figure 9C). We then performed the GO annotation analyses of 392 genes using the PANTHER classification system [56] and zebrafish genome as a reference gene list (Figure 10A,B, Appendix A). The majority of the identified transcripts roughly overlapped with the plausible annotated function: binding, transporter activity, and catalytic activity. Approximately 8% of the genes are related to neurodegenerative disease pathways associated with Alzheimer’s disease, Huntington’s disease, and Parkinson’s disease. Among the most enriched pathways were signaling via serotonin, glutamate, acetylcholine, and histamine neurotransmitters and signaling via hormones that are released from hypothalamic neurons (oxytocin, thyrotropin-releasing hormone, and gonadotropin-releasing hormone). Among the differentially expressed genes, *egr4* and *fosl1a* (both upregulated) and *fosab* and *npas4l* (both downregulated) are isoforms of immediate-early genes that are known to be regulated by calcium signaling [13,24]. We also detected a high level of overexpression of *smc1a*, which encodes a cohesin complex protein that is involved in regulating nervous system development [57]. Some other well-known genes, such as *kcnc1a*, *slc20a2*, *slc25a17*, *slc25a25b*, and *slc31a1*, which encode proteins that are involved in transporter activity, showed significant changes in the expression of *stim2a^−/−^* larvae (Figure 10B). All of these data show that at the level of transcription, *stim2a* deletion leads to an imbalance of processes that are important for brain development and function.

We examined whether the differentially expressed genes belonged to the zebrafish CaTK that was published previously by our group [10]. Surprisingly, the CaTK and our current dataset shared only nine genes (*anxa3a*, *grinab*, *hp*, *hpca*, *mast2*, *pkn3*, *pvalb7*, *slc25a25b*, and *stim2a*). However, in our experimental dataset, we identified the *cdh13*, *scin*, *dgkaa*, *mmp13a*, *LOC101882496*, *LOC103908715*, and *vwa2* genes, which encode proteins that might function in calcium signaling [46], but they were not identified in our earlier work on the zebrafish CaTK [10]. All of the differentially expressed genes in the CaTK are important for neuronal differentiation and function and could be involved in the behavioral phenotype of *stim2a^−/−^* fish.

## 3. Discussion

An important role has been established for STIM proteins in neuronal Ca^2+^ homeostasis and signaling using in vitro mammalian primary neuronal cell cultures (reviewed in [1,9]). However, these molecular pathways are not precisely linked to in vivo data because S*tim2* knockout in mice leads to early lethality before their nervous system is well developed (4–8 weeks of postnatal development) [31,32]. Thus, complex behavior cannot be analyzed in S*tim2* knockout mice [31,32]. In zebrafish, the nervous system develops at an early stage and allows larvae to exhibit complex behavior [42,50,59]. Additionally, because of larval transparency, in vivo Ca^2+^ measurements can be performed in neurons [45,60]. In the present study, we took advantage of these unique features of zebrafish to investigate behavior together with molecular analysis. The zebrafish genome has many duplicated genes, including *stim2a* and *stim2b* [10,45]. Thus, in contrast to early lethality in *Stim2* knockout mouse lines, the knockout of one zebrafish isoform yields a viable animal, which we demonstrated for *stim2a* knockout fish in the present study and *stim2b* knockout fish in our recent study [50].

We showed that *stim2a* knockout caused distinct behavioral changes in zebrafish larvae. We observed hyperactivity in *stim2a^−/−^* zebrafish larvae, which was characterized by a significant increase in mobility. In the open field test, the acceleration and velocity of *stim2a*^−/−^ larvae was higher; they also spend more time moving. We also observed significantly higher thigmotaxis (i.e., a preference for remaining close to the edge of the well) compared with WT zebrafish. Moreover, in *stim2a^−/−^* mutants, we detected the disruption of phototaxis compared with WT zebrafish. The *stim2a^−/−^* larvae reacted to changes in light in the visual–motor response test, thus excluding the possibility of visual deficits in the mutant animals. Lower light preference has been linked to a decrease in the anxiety in zebrafish larvae [50,59,61]. Such unpredictability between phototaxis and thigmotaxis indicates the involvement of different brain regions that control these two processes [51,62]. In addition, it is possible that the brain connectivity network that controls behavior may exhibit differential susceptibility to the loss of stim2 isoforms. We observed higher activity in *stim2a^−/−^* larvae during the low activity phase in the visual–motor response test, indicating that *stim2a^−/−^* mutants may be hyperactive [50,63].

An increase in locomotor activity is often associated with alterations of neurotransmission in the brain, including alterations of glutamate [64] and γ-aminobutyric acid (GABA) [53,65]. Thus, we treated the WT and *stim2a^−/−^* larvae with glutamate and PTZ. The *stim2a^−/−^* zebrafish reacted similarly to both treatments with an increase in activity during all phases of the visual–motor response test compared with the no-treatment condition. Only during the baseline phase, *stim2a*^−/−^ mutants exhibited lower locomotor activity with glutamate treatment compared with WT fish. To link behavior with cellular-signaling events, we next examined Ca^2+^ activity in neurons in vivo. We observed an increase in oscillation frequency but a decrease in the average amplitude of these oscillations in neurons in the optic tectum in *stim2a^−/−^* zebrafish compared with WT fish. It was shown earlier that STIM2 deletion and the subsequent SOCE disruption caused the decrease in oscillations of non-neuronal cells [66,67]. However, our in vivo data showed the increased Ca^2+^ oscillation frequency in brain neurons *stim2a*^−/−^ mutants. The Ca^2+^ oscillation frequency might get compensated due to Stims interaction with other calcium channels, as shown previously, and by others [68,69]. Moreover, Stims 2a and Stim2b have different dissociation constants with Orai, which could affect the Ca^2+^ oscillation frequency [70]. The qPCR analysis of the SOCE component genes showed the significant downregulation of *stim2a*; however, the expression of other SOCE component genes, including *stim2b*, were not affected, indicating that the compensation effect did not occur by the upregulation of *stim2b*. Moreover, the STIM2 interaction with voltage-gated calcium channels (VGCC), α-amino-3-hydroxy-5-methyl-4-isoxazolepropionic acid receptors (AMPA), N-methyl-D-aspartate receptors (NMDA), and plasma membrane Ca^2+^ ATPases (PMCA) might affect the Ca^2+^ oscillation frequency [12,68,71]. In differential genes expression analysis, we focused on CaTK genes (genes that are involved in Ca^2+^ homeostasis and signaling) and transcription regulators. We found many genes that were affected in *stim2a^−/−^* zebrafish, including *anxa3a*, *grinab*, *hp*, *hpca*, *mast2*, *pkn3*, *pvalb7*, and *slc25a25b* (a member of CaTK genes and showed a significant change in expression). Among transcription factors, *fosl1a*, *fosab*, *npas4l*, and *egr4* are isoforms of immediate–early genes that are regulated by Ca^2+^ signaling [13,24]. The exact function of numerous c-*fos* orthologues (e.g., *fosaa*, *fosab*, *fosb*, *fosl1a*, *fosl1b*, *fosl2*, and *fosl2l*) has not been defined in the literature. However, active neurons are known to have higher *c-fos* gene expression as a result of a perturbation of calcium signaling [24,64]. Moreover, based on the published expression patterns of c-*fos* orthologues, we can conclude that they are expressed in the brain [46]. Thus, these genes have differential functions and might be differentially controlled by Ca^2+^ signaling.

The phenotype of the *stim2a* mutant line that we investigated herein is not identical to the phenotypes of another Stim2 mutant (*stim2b*^−/−^) that we reported previously [50]. Both mutants exhibited a hyperactive phenotype, a decrease in phototaxis, and an increase in thigmotaxis. Both lines also exhibited a milder decrease in activity upon turning the light on (Appendix A) [50]. However, responses to PTZ and glutamate treatment were different between the *stim2a* and *stim2b* mutants. For example, the *stim2a^−/−^* line behaved similarly to WT animals with PTZ treatment, whereas the *stim2b^−/−^* line exhibited greater susceptibility to this seizure-inducing drug. With glutamate treatment, the *stim2b^−/−^* line exhibited higher activity during the low activity phase, unlike the *stim2a^−/−^* line, which was less active during the baseline phase but more active during the following low activity phase. Among the notable genes in our RNA sequencing analysis were genes that are potentially involved in distinct features of the *stim2a^−/−^* line, including genes in the CaTK (*grinab*, *hp*, *hpca*, *mast2*, *pkn3*, *pvalb7*, and *slc25a25b*) [10] and some transcription regulators (*egr4*, *fosl1a*, *fosab*, *npas4l*). Notably, the *anxa3a* and *smc1a* genes may be responsible for some similarities in the phenotypes of the *stim2a^−/−^* and *stim2b^−/−^* lines. The *anxa3a* gene encodes the scaffolding protein annexin 3a, which was shown to bind plasma membrane phospholipids in a calcium-dependent manner and regulate calcium oscillations in the human promyeloblast cell line HL-60 [72]. Interestingly, these oscillations were blocked by 2-Aminoethoxydiphenyl borate (2-APB), a selective inhibitor of SOCE, suggesting the involvement of STIM molecules [72]. The *smc1a* gene is a homologue of human *SMC1a* that was impaired in a patient with Cornelia de Lange, a cohesinopathy that is characterized by the early manifestation of intractable epilepsy [57]. Our data suggest diverse roles of Stim2 isoforms in the zebrafish brain that might be involved in the maintenance of neuronal SOCE and the regulation of neuronal activity and genetic programs.

In summary, the present study showed that Stim2a deficiency in neurons led to changes in the levels of numerous mRNAs that encode proteins that are involved in calcium homeostasis. Thus, Stim2a can be considered an indirect regulator of their expression. Together with previous data on *stim2b* knockout zebrafish that were reported recently by our laboratory [50], we were able to conclude that both isoforms of *stim2* affect neuronal activity and zebrafish behavior.

## 4. Materials and Methods

### 4.1. Statement of Ethics

All of the experiments were conducted in accordance with the European Communities Council Directive (63/2010/EEC, 22 September 2010) and performed with permission from the District Veterinary Inspectorate in Warsaw and Ministry of Science and Higher Education issued to the International Institute of Molecular and Cell Biology in Warsaw (registry no. PL14656251 (9 July 2012), 0064 (21 February 2013) and 0051 (31 May 2016), respectively) according to internal regulations of the Zebrafish Core Facility at the International Institute of Molecular and Cell Biology.

### 4.2. Materials

The chemicals and reagents were obtained from the following suppliers: glutamate (Sigma-Aldrich, catalog no. G1251, Saint Louis, MO, USA), pentylenetetrazol (PTZ; Sigma-Aldrich, catalog no. P6500, Saint Louis, MO, USA), pancuronium bromide (Sigma-Aldrich, catalog no. P1918, Saint Louis, MO, USA), low melting agarose (Sigma-Aldrich, catalog no. A9414, Saint Louis, MO, USA), Qiazol (Qiagen, catalog no. 79306, Maryland, USA), the iScript cDNA Synthesis Kit (Bio-Rad, catalog no. 1708890, Hercules, CA, USA), the SuperScript IV First-Strand Synthesis System (Invitrogen, catalog no. 18091050, Carlsbad, CA, USA), Precision Melt Supermix (Bio-Rad, catalog no. 1725112, Hercules, CA, USA), TRI reagent (Invitrogen, catalog no. AM9738, Carlsbad, CA, USA), and RNA Clean and Concentrator Kit (ZYMO Research, catalog no. R1013, Irvine, CA, USA).

### 4.3. Animal Husbandry

All of the experimental animals were maintained in the Zebrafish Core Facility according to standard protocols [73]. The AB zebrafish line was used as the WT control, and the *stim2a^−/−^* mutant line was used as in all of the behavioral tests and the experiments of RNA sequencing and SOCE component expression profiles. The Tg(*HuC:GCaMP5G*) of either *stim2b^−/−^* or *stim2a^−/−^* on the casper background was used for the calcium imaging experiments. Adult fish were kept at a maximum density of 25 fish per tank, with a zebrafish recirculating system and normal photoperiod (14 h/10 h light/dark cycle) at 28.5 °C. Zebrafish eggs for the experiments were obtained by random mating, and each breeding tank was used a maximum of only once weekly. The collected eggs were sorted in E3 water (2.48 mM NaCl, 0.09 mM KCl, 0.164 mM CaCl_2_·2H_2_O, and 0.428 mM MgCl_2_·6H_2_O) at a density of ~50 eggs per 10 cm Petri dish and kept under standard conditions (28.5 °C and normal 14 h/10 h light/dark cycle) until the stage of development was reached for specific experiments. The stages of fish development were defined as hours post-fertilization (hpf) and days post-fertilization (dpf).

### 4.4. Generation and Genotyping of stim2a^−/−^ Mutant Zebrafish Line

Using the clustered regularly interspaced short palindromic repeats (CRISPR)/Cas9 system, the *stim2a^−/−^* zebrafish line was created by introducing a 5 bp deletion before the start of the SOAR domain, which promoted an early stop codon. The fish were genotyped using the RLFP protocol that was developed especially for the *stim2a^−/−^* line. Briefly, at ~30 dpf, tail fin tissue was obtained by biopsy and digested in Tris-EDTA (TE) buffer that was supplemented with proteinase K (0.5 mg/mL final concentration) overnight at 50 °C. The isolated DNA was used for polymerase chain reaction (PCR) using forward (5′-AACTCAGCCGTCTGTGGTATGCG-3′) and reverse (5′-TGACGTTGTAGTACTGAACCTCCACCTC-3′) primers for the *stim2a* gene. The fragments were digested with MBO II restriction enzyme and visualized by electrophoresis in 2% agarose (30 min, 80 mV) with the FastRuler Low Range DNA Ladder. The heavy fragment was designated as belonging to animals with both alleles that carried the *stim2a^−/−^* mutation. The heavy fragment that was accompanied by two light fragments was designated as belonging to animals with only one allele that carried the mutation (i.e., heterozygous). The presence of two light fragments alone was designated as belonging to animals with both alleles that did not carry the mutation (i.e., WT). F2 fish were out-crossed with the AB zebrafish line, and their offspring were in-crossed to obtain homozygous mutants. All of the animals that were used in the experiments were offspring from these fish (or fish from subsequent generations). For the Ca^2+^ imaging experiments, *stim2a^−/−^* fish were out-crossed with Tg(*HuC:GCaMP5G*) and selected for fluorescent homozygous mutants.

### 4.5. Behavioral Experiments

The behavioral experiments were performed according to previously published protocols [38,40,42,50]. The treatment solutions were prepared in E3 water. The final concentrations of the chemicals were the following: 600 μM glutamate (30 mM stock solution) and 15 mM PTZ (0.5 M stock solution). These solutions were mixed in the experimental wells at a 1:1 proportion in E3 water from 2× concentrated working solutions. On the day of the experiment, randomly selected 4 dpf larvae were habituated to the behavioral testing room for at least 30 min. Their locomotor activity was recorded using the ZebraBox high-throughput monitoring system (ViewPoint, Life Sciences, Lyon, France). The video recordings were further analyzed using EthoVision XT software (Noldus, Wageningen, The Netherlands). The data were exported to Microsoft Excel files for further analysis using R software (v.3.6.0, R Foundation for Statistical Computing, Vienna, Austria). Data from larvae that were not active during the entire recording period (total distance < 10 mm) were excluded from the analyses.

#### 4.5.1. Open Field Test

This test was initially developed for rodents [74,75] and subsequently adapted to adult zebrafish and larvae [38,44,75]. The protocol was modified from [38]. Briefly, 2 min before recording in the test, 4 dpf larvae were transferred to a 12-well plate at a density of 1 embryo per well (Figure 3A). The test duration was 15 min. The following parameters were analyzed: total distance traveled in either the border or central zones of the wells, the duration of movements in each area, the duration of not-movement in each area, the mean velocity of the fish, the duration of the movement with a high speed (>20mm/s), and the maximum acceleration. Heat maps showing the mean traces of larvae were generated using EthoVision XT software (Noldus). The 15 min recording period was divided during the data analysis into three phases, 5 min each. Thigmotaxis is presented as the percent time spent in the zones relative to the total time spent moving in the outer zone of the test apparatus [38].

#### 4.5.2. Dark/Light Preference Test

This test was performed according to previous studies [51,61,76,77,78]. A 10 cm Petri dish was used. Half of the Petri dish was covered with a photographic filter, and the side of the dish was covered with black vinyl tape to avoid light penetrance (Figure 3B). The test was recorded for 15 min after 2 min of the adaptation period. The total distance traveled (in millimeters) was analyzed.

#### 4.5.3. Visual–Motor Response Test

This test was described elsewhere [52,79]. Briefly, a 24-well plate was used as the arena (Figure 3C). The fish were acclimatized to the plate before the experiment. The experiments began with an exchange of half the volume of each well into the test solution. The fish were treated with either 600 µM glutamate or PTZ. Untreated fish were exposed to E3 water only. The recording began with a 2 min delay and was divided into the following phases: baseline phase (0% light illumination), low activity phase (70% light illumination), and high activity phase (0% light illumination; Figure 3C). The total distance traveled (in millimeters) was analyzed.

### 4.6. In Vivo Calcium Imaging

The in vivo calcium imaging of neuronal activity in the brain was performed using a Zeiss Lightsheet Z.1 microscope (40×/1.0 objective). Tg(*HuC:GCaMP5G*) [63] *stim2a^−/−^* zebrafish and their *stim2a^+/+^* siblings (WT) were used in the experiment. Zebrafish larvae were immobilized at 4 dpf using 0.6 µg/µl pancuronium bromide and mounted in 1.5% low melting agarose. Time-lapses were recorded for 5 min, with 15 ms exposure and a frequency of 1 frame/sec at single-cell resolution. A plane that contained the habenula, optic tectum, and cerebellum was selected. Automated cell segmentation was done to draw the region of interest (ROI) as described elsewhere [80] in a sub-region in the periventricular gray zone of the optic tectum (as shown in the Appendix A) using MATLAB script (Mathworks). After that, the GCaMP5G fluorescence changes due to spontaneous Ca^2+^ activity were extracted from each cell from the sub-region ROI (Appendix A: plots the raw data of Ca^2+^ fluorescence traces from individual neurons after automated cell segmentation for Appendix A). Then, the peaks of oscillations of Ca^2+^ levels were selected using a function that detects peaks based on a slope change. The average oscillation frequency (in Hz) and the average amplitude (AU) of Ca^2+^ signals were analyzed. The Wilcoxon rank-sum test was used for comparisons of average oscillation frequencies between WT and *stim2a^−/−^* animals. The data are presented as medians with Q1 and Q3 quartiles using boxplots, and dots represent data outliers. The following numbers of cells were analyzed: 168 cells from three WT animals and 392 cells from seven *stim2a^−/−^* animals.

### 4.7. Quantitative PCR Gene Expression Analysis

Thirty zebrafish larvae (5 dpf) per sample were anesthetized with MS-222, collected on ice, washed in phosphate-buffered saline (PBS), and snap-frozen in Qiazol (Qiagen, Hilden, Germany) until use for RNA isolation. RNA was isolated according to a previously published protocol [81]. The RNA template (500 ng) was used for cDNA synthesis using the iScript cDNA Synthesis Kit (Bio-Rad, Hercules, CA, USA). Quantitative PCR (qPCR) was performed in duplicate using 25 ng cDNA per reaction with Precision Melt Supermix (Bio-Rad, Hercules, CA, USA). The analysis of SOCE component gene expression was performed as described elsewhere [10] using previously characterized primers for *stim1a, stim1b, stim2b, orai1a, orai1b, orai2.* Fold changes were calculated using the ΔΔCq method [10]. Eukaryotic translation elongation factor 1 α1, like 1 (*eef1a1l1*) gene expression, was used as the reference gene. Gene expression levels were analyzed using the CFX Connect RT-PCR Detection System (Bio-Rad, Hercules, CA, USA) and visualized using R software.

### 4.8. RNA Sequencing Analysis

The RNA isolation procedure for the RNA sequencing experiment was similar to RNA isolation for the qPCR gene expression analysis [10,81]. Briefly, the total RNA was digested using DNase I and purified using the RNA Clean and Concentrator Kit (ZYMO Research, Irvine, CA, USA) according to the manufacturer’s instructions. Sequencing was performed using Illumina methodology. The preparation of cDNA libraries and sequencing using next-generation sequencing (NGS NextSeq 500, San Diego, CA, USA) were performed in cooperation with the Core Facility at the International Institute of Molecular and Cell Biology in Warsaw. The following sequencing parameters were applied: paired-end sequencing run type, 1 × 76 bp read length. This allowed approximately 120–150 million reads per sample with a 76 bp length. The reads were extracted into FASTQ format and used for the subsequent bioinformatic analysis. FASTQ files are deposited in the Sequence Read Archive (accession no. PRJNA635784). The reads were then aligned to the zebrafish Refseq genome assembly (GRCz11_genomic.fa). Genes were annotated using the GRCz11_genomic.gff Ensembl annotation file. The genes were evaluated according to the false discovery rate (FDR) and fold change between the WT and *stim2a^−/−^* samples. Genes with > 2-fold changes between WT and mutants and a significant FDR were taken for the analysis. The Danio rerio genome was used as a reference gene list, which allowed the identification of cellular components, molecular functions, and the related pathways from the gene ontology (GO) terms [10]. The most promising gene candidates were plotted as the mean ± standard error of the mean (SEM).

### 4.9. Statistical Analysis

The analysis output from EthoVision was exported to Microsoft Excel files and analyzed using R software (v.3.6.0). Results from larvae that were not active during the entire recording period (total distance < 10 mm) were excluded from the analysis. The data distribution was checked using the Shapiro–Wilk normality test. Data from the behavioral experiments were analyzed using paired or unpaired Wilcoxon rank-sum tests for comparisons between the groups. Adjusted values of *p* < 0.05 were considered statistically significant. The data are expressed as medians with Q1 and Q3 quartiles using boxplots, and dots represent data outliers unless stated otherwise. The number of fish that were used in each experiment is indicated in the figure legends. The RNA sequencing data were analyzed based on logarithmic (log_2_) fold changes and are expressed as mean ± SEM if the log_2_ fold change was >2 and <−2. The FDR was used to estimate the statistical significance of gene expression using RNA sequencing.

## Figures and Tables

**Figure 1 ijms-21-06198-f001:**
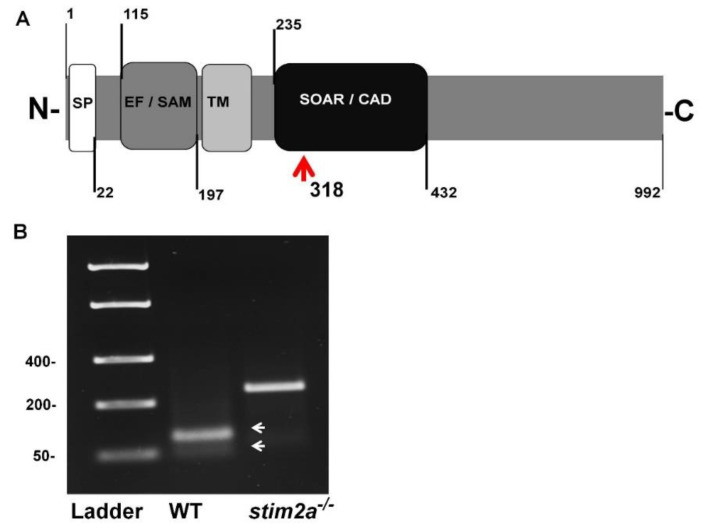
Generation of *stim2a^−/−^* mutant line. (**A**) Schematic illustration of Stim2a protein domains. SP, signal peptide; EF/SAM, EF-hand/sterile α-motif (EF/SAM) Ca^2+^-binding endoplasmic reticulum (ER)-luminal domain; TM, transmembrane domain; SOAR/CAD, STIM1 Orai1-activating region/CRAC-activating domain. Numbers indicate amino acid positions. The red arrow indicates the site of the stop codon introduction. (**B**) Representative gel electrophoresis of the restriction length fragment polymorphism (RLFP) analysis of genotypes (white arrows indicate wildtype (WT) fragments after the cutting) of the PCR product. The mutant is not cleaved.

**Figure 2 ijms-21-06198-f002:**
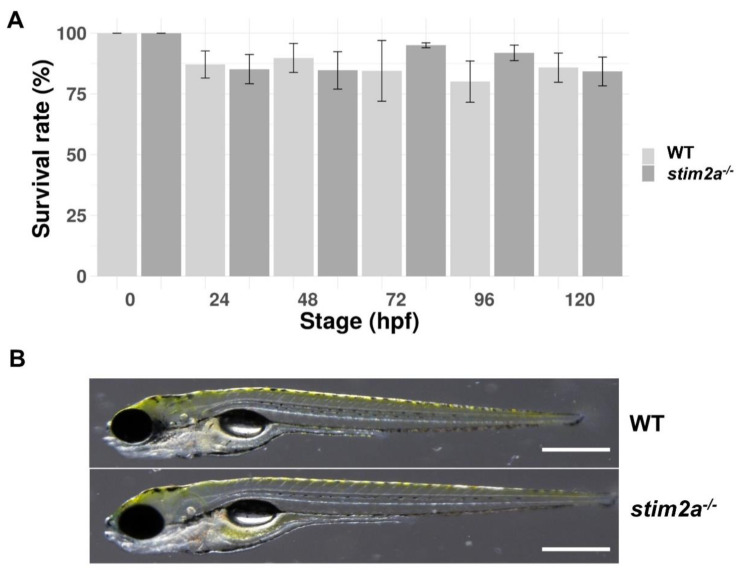
The survival and morphology of *stim2a^−/−^* zebrafish did not differ from WT animals. (**A**) Percentage of surviving WT and *stim2a^−/−^* zebrafish during the first 120 h of development. The data are expressed as the mean ± SEM. Number of repetitions: 3. Number of larvae for each repetition: 200. (**B**) Side views of *stim2a^−/−^* and WT zebrafish at 5 dpf. Scale bars = 500 µm.

**Figure 3 ijms-21-06198-f003:**
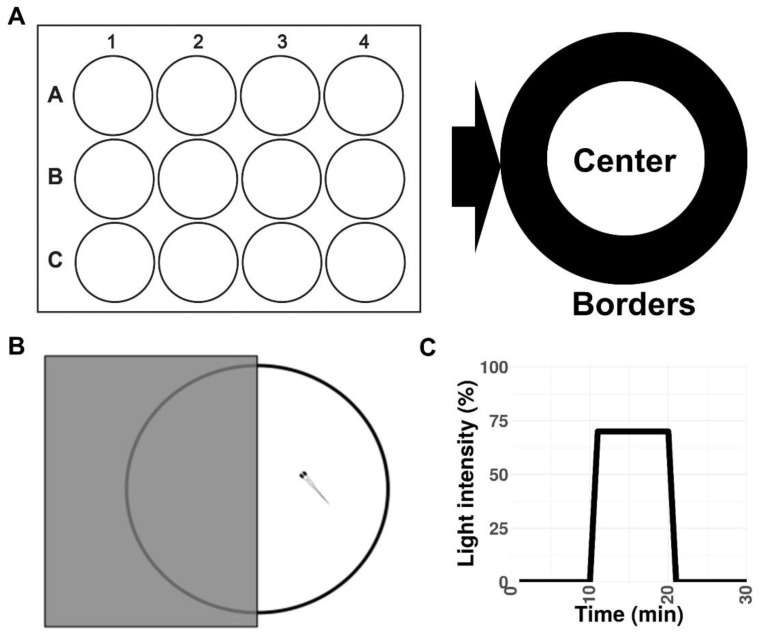
Schematic representations of the experimental setup. (**A**) Open field test in a 12-well plate. (**B**) Light/dark preference test in a 10 cm Petri dish. (**C**) Visual–motor response protocol.

**Figure 4 ijms-21-06198-f004:**
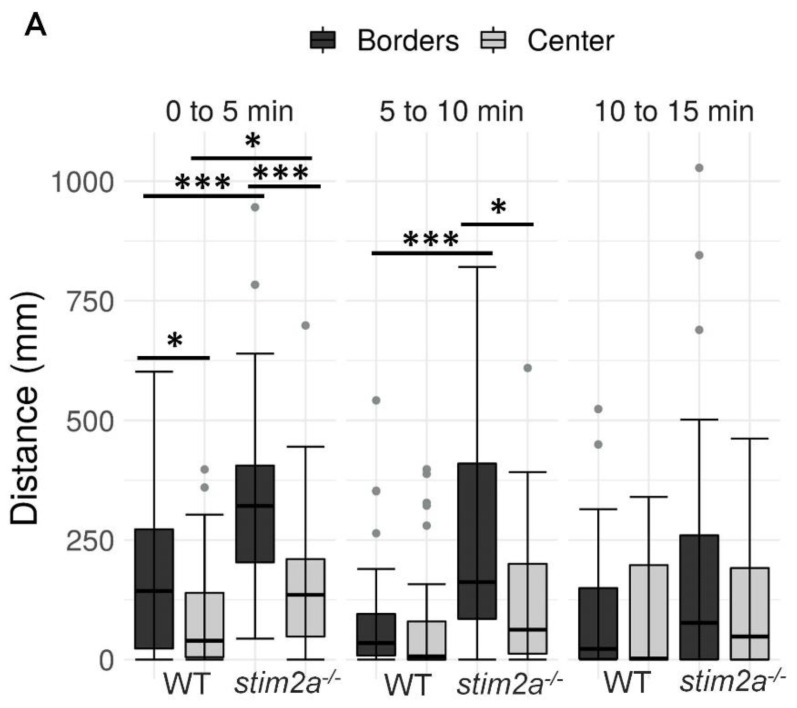
Higher thigmotaxis in *stim2a^−/−^* zebrafish in the open field test adopted for zebrafish larvae. Activity of the larvae was recorded for 15 min in a 12-well plate. The experiment was divided into three phases, 5 min each (0–5 min, 5–10 min, and 10–15 min). Fish that did not move were not included in the analysis. (**A**) Box plots show the distance traveled (in millimeters) in WT and *stim2a*^−/−^ fish at either the edge of the well or in the center. (**B**) Box plots of the time spent moving (duration of movement) while swimming either at the edge of the well or in the center. (**C**) Box plots show the maximum acceleration (in millimeters per squared second) of WT and *stim2a*^−/−^ fish in total throughout the well. (**D**) Heat maps show the total distance covered by the larvae in each phase of the experiment. * *p* < 0.05, *** *p* < 0.001 (paired Wilcoxon rank-sum test for comparisons between the edge and center; unpaired Wilcoxon rank-sum test for comparisons between WT and *stim2a*^−/−^ mutants). *n* = 32 for WT larvae. *n* = 33 *stim2a*^−/−^ larvae. Number of experiments: 3.

**Figure 5 ijms-21-06198-f005:**
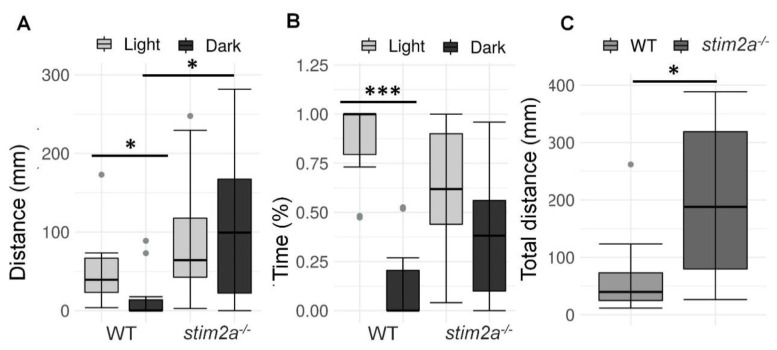
Hyperactivity and lower phototaxis in *stim2a*^−/−^ fish in the light/dark preference test. This test was performed for 15 min, during which half of the Petri dish was covered with a photographic filter that blocked the light. (**A**) Box plots of the distance traveled (in millimeters) in the light and dark zones of the Petri dish in the WT and *stim2a*^−/−^ larvae. (**B**) Box plots of the time spent moving (duration of movement) in the light and dark zones in WT and *stim2a*^−/−^ larvae. (**C**) Box plots of the cumulative distance traveled (in millimeters) during the experiment by the WT and *stim2a**^−/−^* larvae. * *p* < 0.05, *** *p* < 0.001 (paired Wilcoxon rank-sum test for comparisons between the light and dark zones; unpaired Wilcoxon rank-sum test for comparisons between WT and *stim2a*^−/−^ larvae). *n* = 10 WT larvae. *n* = 11 *stim2a*^−/−^ larvae. Number of experiments: 5.

**Figure 6 ijms-21-06198-f006:**
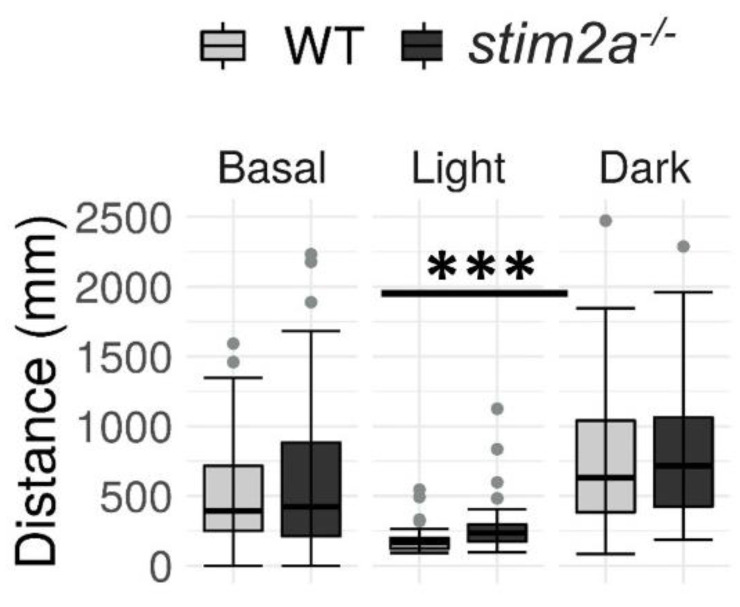
The *stim2a*^−/−^ larvae reacted to changes in the light and exhibited hyperactivity during the low activity phase in the visual–motor response test. Activity was recorded during a 30 min period that consisted of a baseline phase (basal, 0% light illumination), low activity phase (light; 70% light illumination), and high activity phase (dark; 0% light illumination). Box plots show the distance traveled (in millimeters) during the respective phase. *** *p* < 0.001 (Wilcoxon rank-sum test). *n* = 53 WT larvae. *n* = 52 *stim2a*^−/−^ larvae. Number of experiments: 6.

**Figure 7 ijms-21-06198-f007:**
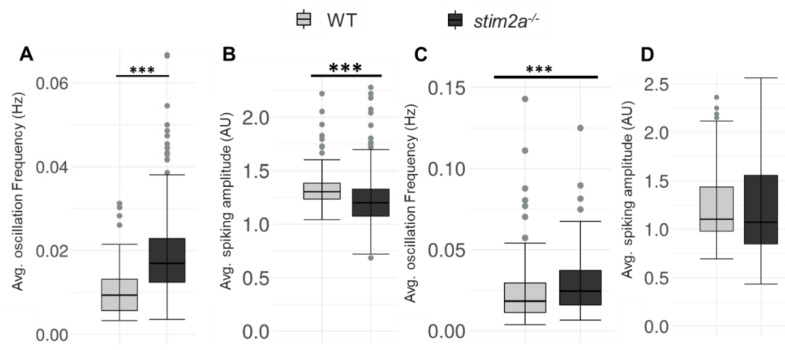
Increase in the spike frequency of neurons in stim2a-deficient zebrafish. For in vivo Ca^2+^ imaging, WT and *stim2a^−/−^* zebrafish that expressed GCaMP5G under the pan-neuronal HuC promoter were used. Zebrafish neurons from the periventricular gray zone of the optic tectum were analyzed. (**A**) Box plots show the basal level (in E3 water) average oscillation frequency (in Hz). (**B**) The basal level (in E3 water) average calcium oscillation amplitude (AU). Number of cells: 168 WT, 392 *stim2a^−/−^* (**C**) Box plots show the average oscillation frequency (in Hz) after treatment with 600 μM glutamate. (**D**) Average calcium oscillation amplitude (AU) after treatment with 600 μM glutamate. Number of cells: 276 WT, 166 *stim2a^−/−^*. The Wilcoxon rank-sum test was performed to compare in vivo Ca^2+^ responses of the brain between WT and *stim2a^−/−^* animals. Number of animals: 3 WT, 7 *stim2a^−/−^*. *** *p* < 0.001.

**Figure 8 ijms-21-06198-f008:**
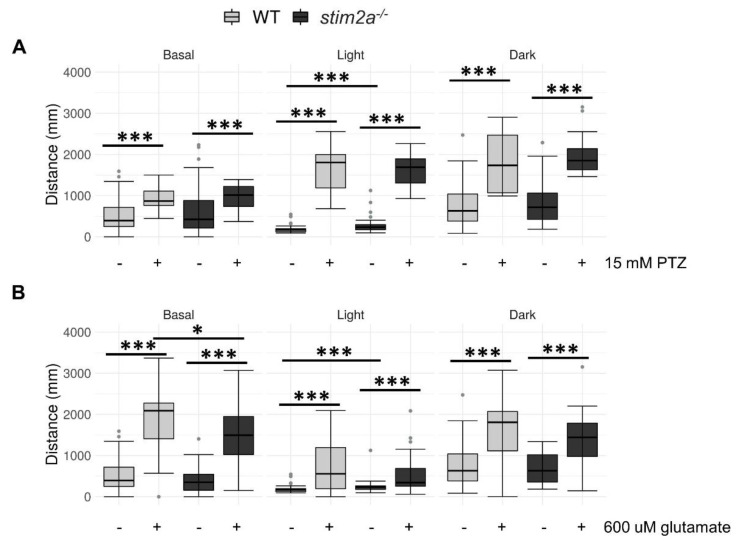
Exposure to Pentylenetetrazol (PTZ) and glutamate did not induce additional hyperactivity in *stim2a^−/−^* larvae. (**A**) Box plots of the distance traveled (in millimeters) in WT and *stim2a*^−/−^ larvae that were treated with PTZ. (**B**) Box plots of the distance traveled (in millimeters) in WT and *stim2a*^−/−^ larvae that were treated with glutamate. Activity was recorded during a 30 min period that consisted of a baseline phase (basal, 0% light illumination), low activity phase (light, 70% light illumination), and high activity phase (dark, 0% light illumination). Before the experiment, half of the medium was exchanged for either PTZ solution (15 mM final concentration) or glutamate solution (600 μM). –, untreated; +, treated. * *p* < 0.05, *** *p* < 0.001 (Wilcoxon rank-sum test). *n* = 18 larvae/group for the PTZ treatment. *n* = 36 WT larvae and *n* = 35 for *stim2a^−/−^* larvae for the glutamate treatment. Number of experiments: 3.

**Figure 9 ijms-21-06198-f009:**
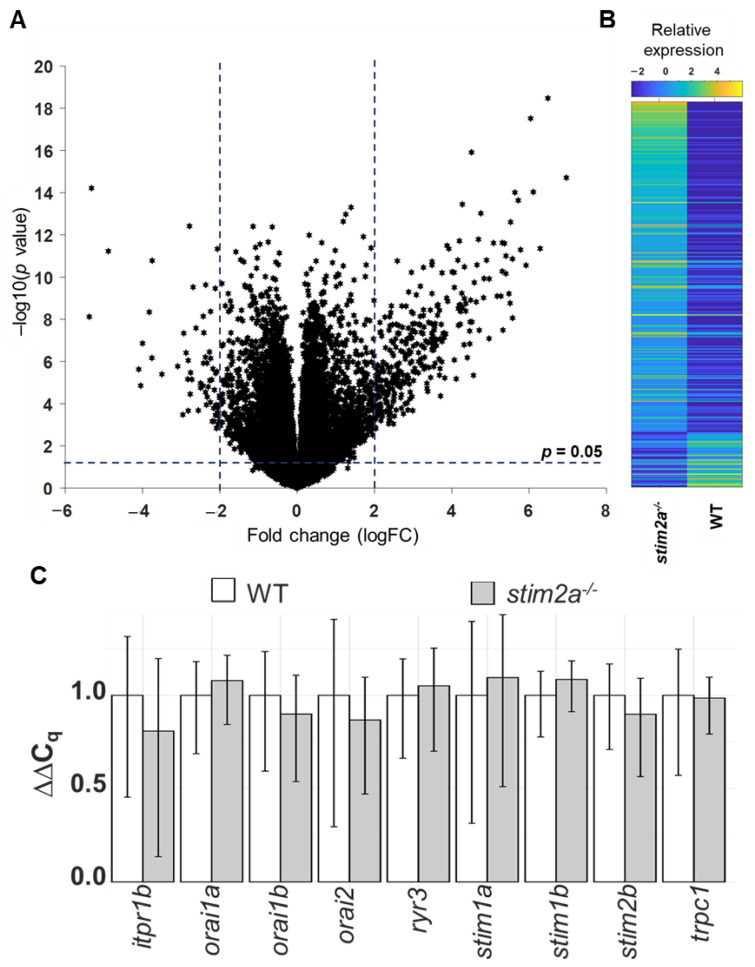
RNA sequencing analysis. (**A**) Volcano plot of the transcriptional differences between the *stim2a^−/−^* and WT zebrafish larvae. The logarithms of the fold changes in the individual genes (*x* axis) are plotted against negative logarithms of their *p*-value to base 10 (*y* axis). Positive log_2_ (fold change) values represent the upregulation in *stim2a^−/−^* larvae compared with WT larvae, and negative log_2_ (fold change) values represent downregulation. Points above the dotted line represent differentially expressed genes in *stim2a^−/−^* larvae with *p* < 0.05 after correction for multiple testing. (**B**) Heat map of a total of 392 genes that were identified as being differentially expressed (log_2_ (fold change) ≥ 2) between the *stim2a^−/−^* and WT larvae, including 336 upregulated genes and 56 downregulated genes. (**C**) mRNA levels of store-operated calcium entry (SOCE) components in 5 dpf zebrafish, quantified by qPCR.

**Figure 10 ijms-21-06198-f010:**
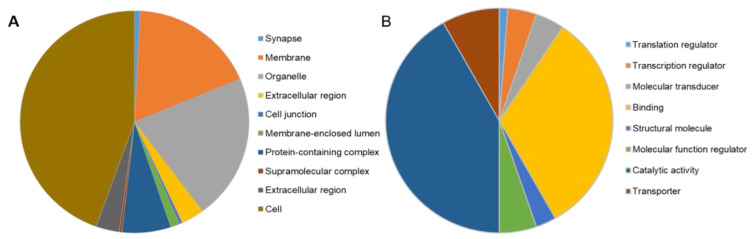
PANTHER Gene Ontology annotation analysis [58]. The distribution of gene ontology (GO) terms was categorized based on PANTHER GO-Slim. (**A**) Cellular Component GO terms (components of cells or extracellular) with 538 component hits. (**B**) Molecular Function GO terms (basic activities of a gene product at the molecular level, such as binding or catalysis) with 170 function hits.

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
