# Peer review of "Knockout of stim2a Increases Calcium Oscillations in Neurons and Induces Hyperactive-Like Phenotype in Zebrafish Larvae"

_ijms, 2020, doi:10.3390/ijms21176198_

Round 1

Reviewer 1 Report

In this manuscript the authors described a hyperactive-like phenotype in stim2a KO zebrafish larvae. Specifically they detected an increase in activity during the visual motor response test, an increase in 
thigmotaxis in the open field test, and a disruption of phototaxis in the dark/light preference test in stim2a KO mutants compared with WT.

I found the behavioral experiments well done. However I have major concerns for which I do not recommend the publication of this work, at least in the present form.

This work overlaps perfectly with the previous work from the same authors published only one month ago in Cells: ‘stim2b Knockout Induces Hyperactivity and Susceptibility to Seizures in Zebrafish Larvae’. They performed the same behavioral tests, analyzed the same brain regions, studied the same Ca2+ dynamics (frequency of Ca2+oscillations) and performed the same gene expression analysis. The relevance of that work is that it was original and stim2b, in contrast to stim2a, is highly expressed in the brain of zebrafish larvae. In fact, the absence of stim2b induces a drastic phenotype in zebrafish larvae.

Both factors (duplication of the previous work and low expression levels of stim2a in the zebrafish brain with associated mild phonotype of stim2a KO zebrafish) weaken substantially either the novelty or the scientific relevance of this work. Indeed, in my opinion, this paper can be suitable for publication only if new insights on the role of stim2 in the zebrafish neurons are added.

For instance more detailed studies on Ca2+ dynamics must be done in brain of stim2a KO zebrafish, such as the study of the shape/amplitude of Ca2+ spikes (not only the frequency), the analysis of Ca2+ levels in the ER (using ERD1 Ca2+probe for instance). Finally the effect of glutamate on Ca2+ response needs to be investigated in these neurons. The authors did not even discuss the possible link between the increase in the Ca2+ spike frequency observed in the neurons of stim2a KO larvae and the possible absence of a proper SOCE and ER refill in these neurons. How a possible decrease in Ca2+ entry due to the absence of stim2a may induce an increase in Ca2+ oscillation frequency? What about the role of major stim2b isoform in the authors’ model in which they selectively knocked out only stim2a? What about the morphology of the neurons of stim2a KO larvae?

These questions require answers, in my opinion, to consider this study a step forward compared to the previous one from the same group.

Author Response

We thank the Reviewer for careful analysis of our manuscript and valuable remarks. We addressed all of them, including adding data of additional experiments. We believe that the comments from the Reviewer helped us in improving the manuscript to a great extent. The changes in the text in the revised version of the manuscript are marked using the "Track Changes" function.

Reviewer 2 Report

The manuscript by Palchevska and colleagues describes the phenotype of a zebrafish stim2a mutant. The analysis is carefully done, albeit in the end a little inconclusive.

Minor technical points:

For the phototaxis experiment, it is important to note the adaptation state of the larvae. Phototactic behavior is strongly dependent on the light adaptative state of the larvae.

Although the expression pattern in the mouse is reported in the introduction, nothing is mentioned about the expression pattern in the zebrafish. This is particularly important since two paralogues of this gene exist. With this manuscript we have the phenotypic description of mutant strains in both paralogues, it is most useful to compare this with the respective expression of these genes.

Other comments:

In section 2.2 the authors need to state the age of the animals. It reads as if the authors analyzed adult fish.

Why did the authors restrict their analysis to the larval stage. Adults are reported to be viable and fertile. It would be straight forward to assess e.g the swimming behavior of the adults.

The brain imaging data is interesting but unnecessary restricted. The authors should at least state why on the tectum was analyzed. An obvious measurement would be to compare spontaneous whole-brain neuronal activity. Oscillatory activity is interesting, but what about general activity. I do not understand the sentence "..optic tectum neurons fired much faster ...". Neurons fire always with the same speed, but the frequency may differ (action potentials are "all or nothing" events). Do you mean with higher frequency?

Given the higher frequency of activity, the PTZ experiment is inconclusive. Why didn't the authors use concentrations of PTZ that induce epileptic seizures? Then they could ask the important question if the stim2s mutant brain is more prone to seizures.

The RNA sequencing analysis is superficial and the conclusion unfounded. Why is stim2a message in the mutant? If the differentially expressed genes are really involved in neuronal differentiation, the phenotype would be much stronger.

Through the manuscript, the obvious question of compensation is not addressed. Since the zebrafish has two paralogues that may only partially subfunctionalized, the mild phenotype may very well be explained by compensation via stim2b, which is apparent, not upregulated. Has the double mutant a stronger phenotype?

Since now this report the phenotype of both paralog

The interpretation of the phenotype is hampered

Author Response

We thank the Reviewer for careful analysis of our manuscript and valuable remarks. We addressed all of them, including adding data of additional experiments, We believe that the comments from the Reviewer helped us in improving the manuscript to a great extent. The changes in the text in the revised version of the manuscript are marked using the "Track Changes" function.

Reviewer 3 Report

Review of Palchevska et al,

Palchevska et al. described that stim2a KO zebrafish showed dis-regulation of calcium homeostasis and hyper-active phenotype.  Stromal interaction molecule (STIM) proteins are necessary for the store-operated calcium entry (SOCE) as endoplasmic reticulum Ca2+ sensors.   Both STIM1and STIM2 is expressed, but the function of STIM proteins was not well understood in neurons. Therefore, the authors analyzed behavior, neural activity, and transcriptome in zebrafish (Danio rerio) that lacked stim2a.  The authors finally concluded that the hyperactive-like phenotype of stim2a-/- mutant zebrafish is caused by the dysregulation of Ca2+ homeostasis and signaling.

Behavioral tests, including open field test, light/dark preference test, and visual-motor response exhibited hyperactive phenotype in stim2a-/- mutant zebrafish.  However,

it seems that the locomotor activity is slightly increased rather than hyper-active in tim2a-/- mutant fish. Even the most obvious difference showed in figure 1A, I would suggest that the mutant fish showed simply an increase in locomotor activity. I could not understand the hyperactive phenotype in mutant fish.

The authors observed an increase in oscillation frequency in neurons in the optic tectum in stim2a-/- zebrafish compared with WT fish.  However, the movie (S1) is not a single-cell resolution; it is difficult to calculate the spike frequency of neurons at single-cell resolution.  Besides, the authors recorded the neurons from the periventricular gray zone of the optic tectum, although they did not show any region of interest (ROI). Thus, it is difficult to identify which neurons the authors recorded form the periventricular gray zone in stim2a-/- zebrafish.

The authors performed RNA sequencing in stim2a-/- zebrafish compared with WT fish, and identified genes that are involved in Ca2+ homeostasis and signaling were differentially expressed in the mutant fish.  They identified the 392 genes as being differentially expressed; they did not show how specific alteration occurs in the genes responsible for Ca2+ homeostasis. Besides, mRNA levels of SOCE components are normal in stim2a-/- zebrafish. Thus, there is no evidence of whether knockout of stim2a disturbs calcium homeostasis. 

The work of Palchevska et al. has serious flaws. Therefore, I cannot recommend this article for publication.

Author Response

(The authors gave the same response as above.)

Round 2

Reviewer 2 Report

I am happy with all the changes that the authors have done.

Reviewer 3 Report

Review of Gupta et al.,

Unfortunately, the authors did not satisfactorily answer my comments and questions. Therefore, I do not support the publication.

#1 Behavioral tests

The authors mentioned that "hyperactive phenotype" is applicable (especially Fig. 5C is showing about 4X higher median of total distance in mutants).

However, the authors analyzed just total distance traveled even in Fig. 5C. Thus, the results can be explained as "increasing locomotor activity" but not "hyperactive phenotype." If the authors want to say "hyperactive phenotype," they need to quantify the "hyperactive phenotype." For example, they would need to quantify acceleration (rate of change of velocity) and swimming tracks upon light, at least.  

#2 Oscillation frequency in the optic tectum

The movies (S1 and 2) are the same data, and the authors have not performed new experiments. Again, the movies are not a single-cell resolution. The authors have briefly explained the methodology to draw the ROI, but not described which areas of the neurons they recorded form the periventricular gray zone in stim2a-/- zebrafish.  

Besides, I do not understand the importance of Ca2+ signal oscillations of the optic tectum in this study. Probably, the authors analyzed the behavioral response to light stimulus, and the tectum is known to mediate light response. Therefore, they chose the tectum as suitable brain regions to recorded neuronal activity. However, the authors have only recorded spontaneous neuronal activity in the tectum. The authors should test whether Ca2+ signal oscillations change with or without light stimulus like their behavioral experiments. Thus, it is still challenging to understand causality between behavior response to light and the spontaneous neuronal activity in the tectum. 

#3 RNA sequencing

As I pointed out, there is no evidence of whether knockout of stim2a disturbs calcium homeostasis. The title is "Knockout of stim2a disturbs calcium homeostasis in neurons~", I do not understand the meaning.

Again, I do not support the publication.

Author Response

We thank the Reviewer for careful analysis of our manuscript and valuable remarks. We addressed all of them, including adding data of additional experiments. We believe that the comments from the Reviewer helped us in improving the manuscript to a great extent. The changes in the text in the revised version of the manuscript are marked using the "Track Changes" function.

#1 Behavioral tests

The authors mentioned that "hyperactive phenotype" is applicable (especially Fig. 5C is showing about 4X higher median of total distance in mutants).

However, the authors analyzed just total distance traveled even in Fig. 5C. Thus, the results can be explained as "increasing locomotor activity" but not "hyperactive phenotype." If the authors want to say "hyperactive phenotype," they need to quantify the "hyperactive phenotype." For example, they would need to quantify acceleration (rate of change of velocity) and swimming tracks upon light, at least.  

Response:

We propose hyperactive because we observe consistently increased distance traveled by stim2a-/- larvae in several behavioral tests we used (what was especially visible in Fig. 5C is showing about 4X higher median of total distance in mutants). Following the Reviewer suggestion, we included in the revised version of the manuscript also previously not shown parameters of mutants activity such as: maximum acceleration (Fig. 4C), time spend moving (Supplementary Fig. S1A), velocity (Supplementary Fig. S1B), and time spent moving with a high speed (Supplementary Fig. S1C). All those parameters were increased in mutants, which further confirm it is a hyperactive phenotype. We also have included heat maps showing mean traces of WT and stim2a-/- larvae (Fig. 4D). We chose an open field test to perform those analyses since we are able to observe the spontaneous behavior of larvae, that is not affected by any other factors like light or drug treatment. We referred to the new results in the revised version of the manuscript in the Results, Discussion and Methods sections in lines 146-152, 165-168, 343-345, and 475-479 using “Track Changes” option.

#2 Oscillation frequency in the optic tectum

The movies (S1 and 2) are the same data, and the authors have not performed new experiments. Again, the movies are not a single-cell resolution. The authors have briefly explained the methodology to draw the ROI, but not described which areas of the neurons they recorded form the periventricular gray zone in stim2a-/- zebrafish.  

Besides, I do not understand the importance of Ca2+ signal oscillations of the optic tectum in this study. Probably, the authors analyzed the behavioral response to light stimulus, and the tectum is known to mediate light response. Therefore, they chose the tectum as suitable brain regions to recorded neuronal activity. However, the authors have only recorded spontaneous neuronal activity in the tectum. The authors should test whether Ca2+ signal oscillations change with or without light stimulus like their behavioral experiments. Thus, it is still challenging to understand causality between behavior response to light and the spontaneous neuronal activity in the tectum. 

Response:

We thank the Reviewer for rising above concerns. We have gone through the representative videos (Supplementary video S1 and S2; controls: WT and stim2a-/- in E3 medium) and confirm they are correct. We have performed new experiments and added two representative videos for the glutamate treatment (Supplementary video S3 and S4 for WT and stim2a-/- respectively). As the experimental conditions are the same for all recordings, movies look almost the same. We have provided an additional supplementary figure (Supplementary Fig. S3A-D), which plots the raw data of Ca fluorescence traces from individual neurons after automated cell segmentation for Supplementary Video S1, S2, S3, and S4. We referred to the above modifications in the revised version of the manuscript in the Results and Methods sections in lines 223-237 and 500-510 using the “Track Changes” option.

Regarding the single-cell resolution issue, we believe that it is what we show in the paper. This is based on the instance on the paper by (Panier, Romano et al. 2013) (Fig. 2A and 2E). They performed imaging on zebrafish larvae expressing the genetically encoded calcium indicator GCaMP3, using Selective-plane Illumination Microscopy (SPIM), illuminated with a scanned laser sheet and imaged with a camera whose optical axis was oriented orthogonally to the illumination plane. Furthermore, Panier et al. showed that the optical transparency of zebrafish larvae and optical sectioning approach permits functional imaging of a very large fraction of the brain volume with single- or near single-cell resolution. We use the same methodology for the experiments of our manuscript. For the imaging and recording, we use the plane that contained the habenula, optic tectum, and cerebellum. Then automated cell segmentation on sub-region in the periventricular gray zone of the optic tectum (as shown in Supplementary Fig. S5) was performed using MATLAB script. Accordingly, we have modified the texts in the revised version of the manuscript in the Methods sections in lines 500-509 using the “Track Changes” option.

We would like to explain why the periventricular gray zone of the optic tectum was chosen to analyze the spontaneous Ca2+ activity. This analysis could be performed in another place of the zebrafish brain since we recorded the activity of all neurons visible in the whole plane. This area was chosen because it was showing intensive oscillations in all analyzed zebrafish lines and was easily identified in each experiment to allow comparisons between them. Analysis of spontaneous activity in the optic tectum was performed not to analyze the effect of light, which would be very interesting, compatible with some behavior tests, but to see if the lack of Stim2a has any effect. We believe that we achieved this goal since we detected changes in calcium oscillations. We have modified the text in the revised manuscript in the Results section to provide precise readability. Modified texts in lines 226-229 are marked using the "Track Changes" function in the revised manuscript.

#3 RNA sequencing

As I pointed out, there is no evidence of whether knockout of stim2a disturbs calcium homeostasis. The title is "Knockout of stim2a disturbs calcium homeostasis in neurons~", I do not understand the meaning.

Response:

We agree with the Reviewer that the earlier title was not appropriate and so we considered changing the title. The  new title is “Knockout of stim2a increases calcium oscillations in neurons induces hyperactive-like phenotype in zebrafish larvae.”

Reference:

Panier, T., S. A. Romano, R. Olive, T. Pietri, G. Sumbre, R. Candelier and G. Debregeas (2013). "Fast functional imaging of multiple brain regions in intact zebrafish larvae using selective plane illumination microscopy." Front Neural Circuits 7: 65.

Round 3

Reviewer 3 Report

The authors satisfactorily answered my comments and questions.

I am satisfied by the description of the hyperactive phenotype. I appreciate the precisions they added as well as additional experiments to test the behavioral phenotype.

There are many questions open now for the field to investigate in particular regarding the Ca2+ oscillation and zebrafish behavior.